# The Diagnostic Utility of Artificial Intelligence-Guided Computed Tomography-Based Severity Scores for Predicting Short-Term Clinical Outcomes in Adults with COVID-19 Pneumonia

**DOI:** 10.3390/jcm12227039

**Published:** 2023-11-10

**Authors:** Zeynep Atceken, Yeliz Celik, Cetin Atasoy, Yüksel Peker

**Affiliations:** 1Department of Radiology, Koc University School of Medicine, Istanbul 34010, Turkey; zatceken@kuh.ku.edu.tr (Z.A.); catasoy@kuh.ku.edu.tr (C.A.); 2Center for Translational Medicine (KUTTAM), Department of Pulmonary Medicine, Koc University School of Medicine, and Koc University Research, Koc University, Istanbul 34010, Turkey; yecelik@ku.edu.tr; 3Department of Molecular and Clinical Medicine, Sahlgrenska Academy, University of Gothenburg, 40530 Gothenburg, Sweden; 4Department of Clinical Sciences, Respiratory Medicine and Allergology, Faculty of Medicine, Lund University, 22185 Lund, Sweden; 5Division of Pulmonary, Allergy, and Critical Care Medicine, University of Pittsburgh School of Medicine, Pittsburgh, PA 15213, USA; 6Division of Sleep and Circadian Disorders, Harvard Medical School, Brigham and Women’s Hospital, Boston, MA 02115, USA

**Keywords:** COVID-19, artificial intelligence, chest CT, clinical outcomes, supplemental oxygen

## Abstract

Chest computed tomography (CT) imaging with the use of an artificial intelligence (AI) analysis program has been helpful for the rapid evaluation of large numbers of patients during the COVID-19 pandemic. We have previously demonstrated that adults with COVID-19 infection with high-risk obstructive sleep apnea (OSA) have poorer clinical outcomes than COVID-19 patients with low-risk OSA. In the current secondary analysis, we evaluated the association of AI-guided CT-based severity scores (SSs) with short-term outcomes in the same cohort. In total, 221 patients (mean age of 52.6 ± 15.6 years, 59% men) with eligible chest CT images from March to May 2020 were included. The AI program scanned the CT images in 3D, and the algorithm measured volumes of lobes and lungs as well as high-opacity areas, including ground glass and consolidation. An SS was defined as the ratio of the volume of high-opacity areas to that of the total lung volume. The primary outcome was the need for supplemental oxygen and hospitalization over 28 days. A receiver operating characteristic (ROC) curve analysis of the association between an SS and the need for supplemental oxygen revealed a cut-off score of 2.65 on the CT images, with a sensitivity of 81% and a specificity of 56%. In a multivariate logistic regression model, an SS > 2.65 predicted the need for supplemental oxygen, with an odds ratio (OR) of 3.98 (95% confidence interval (CI) 1.80–8.79; *p* < 0.001), and hospitalization, with an OR of 2.40 (95% CI 1.23–4.71; *p* = 0.011), adjusted for age, sex, body mass index, diabetes, hypertension, and coronary artery disease. We conclude that AI-guided CT-based SSs can be used for predicting the need for supplemental oxygen and hospitalization in patients with COVID-19 pneumonia.

## 1. Introduction

The novel coronavirus disease (COVID-19) epidemic has severely affected the lives of millions of people, causing a major public health crisis worldwide. According to the World Health Organization, as of 16 October 2023 COVID-19 has affected more than 770 million people worldwide, with almost 7 million deaths [1]. Due to the primary involvement of the respiratory system, chest CT imaging has been useful in the diagnosis, initial evaluation, and follow-up of COVID-19 patients during the pandemic [2,3]. Determining the severity of COVID-19 pneumonia is crucial, particularly at the first referral for predicting the need for hospitalization, supplemental oxygen, and the intensive care unit (ICU) ward. Optimizing the initial triage of patients could help rationalize the clinical management and cost-beneficial use of hospital resources in order to decrease the adverse health impacts of the disease [3,4,5,6,7,8,9,10,11,12,13].

The severity of lung involvement caused by pneumonia can be quantified via chest computed tomography (CT) [3]. Several scoring systems have been used to standardize the assessment and reporting of COVID-19 on non-contrast chest CT [14,15]. These severity scoring systems have been helpful in making comparative evaluations of the disease and predicting prognoses [8,12,16]. There are three main scoring systems for CT severity evaluation: semiquantitative (visual), quantitative, and artificial intelligence (AI) scoring systems. The AI analysis program scans CT images in 3D and provides the disease-related opacity percentage, high-opacity percentage, lung severity score, and high-opacity score. These three systems have their own advantages and disadvantages. Quantitative scoring has been suggested to be superior to that of semiquantitative analysis [17], which is time-consuming and prone to interobserver variability [18].

In labor-intensive situations such as the COVID-19 pandemic, it is essential to determine the degree of a disease accurately and in a short time. AI systems may enable the rapid evaluation of a large number of images, the assessment of disease severity, the estimation of prognoses, and the evaluation of treatment responses [19]. In a previous study on 279 patients, the AI system achieved an area under the curve (AUC) of 0.92 and had equal sensitivity as compared to a senior thoracic radiologist [20] Interestingly, the AI system also advanced the detection of patients who tested positive for COVID-19 via polymerase chain reaction (PCR) and presented with normal CT scans, correctly recognizing 17 of 25 (68%) patients, whereas radiologists categorized all of these patients as COVID-19-negative [20]. It should also be emphasized that the incorporation of radiology scans in conjunction with laboratory parameters is of supreme importance in the diagnostic and initial clinical assessment process [21]. Thus, the synergy between radiology and laboratory data contributes to a more holistic and effective diagnostic approach, potentially improving patient outcomes and clinical decision making.

The aim of this study was to evaluate the relationship between AI-derived severity scores obtained from CT scans and short-term clinical outcomes.

## 2. Materials and Methods

### 2.1. Study Population

This study is a post hoc analysis of a double-center, prospective evaluation of a COVID-19 cohort during the first outbreak of the pandemic between March and May 2020 [22]. All COVID-19 cases were confirmed via the PCR testing of nasopharyngeal specimens and/or clinical symptoms and radiologic findings suggestive of COVID-19 pneumonia [22]. As shown in Figure 1, 320 out of 472 consecutive adults who were diagnosed with COVID-19 were eligible for the main study. The participants of the current protocol were from two hospitals (Koç University Hospital and Koç Healthcare American Hospital, Istanbul, Turkey), admitted to the hospitals from central districts of Istanbul between 10 March and 22 June 2020. For the current protocol, it was decided that pneumonic scoring would not be appropriate in patients with primary and metastatic lung malignancies, tuberculosis with dense fibrotic scars, and large atelectasis, since total aerated lung parenchyma would decrease and pneumonic infiltration opacities would be indistinguishable from the opacities of the underlying disease, and thus would lead to inaccurate results. Thus, the inclusion criteria were being included in the main study and having eligible CT scans for an AI analysis. Exclusion criteria were the lack of an AI analysis as well as concomitant primary or secondary lung malignancies or a history of tuberculosis. As illustrated in Figure 1, patients with these comorbidities (*n* = 23), as well as ones with non-eligible chest CTs (*n* = 76), were excluded from the analysis.

The study protocol was approved by the Koç University Committee on Human Research (approval nr 2020.140.IRB1.030; 04.15.2020), and written informed consent was provided from all of the patients. The trial was registered with the ClinicalTrials.gov (NCT04363333).

### 2.2. Chest MDCT Protocol and Assessment

All patients were scanned with a 64-detector row CT scanner (Somatom^®^ Definition AS; Siemens Healthineers, Forchheim, Germany). Scanning was performed in the supine position and after full inspiratory breath-hold. The scanning range was from the lung apices to the costophrenic angles. Low-radiation-dose protocols were utilized, and X-ray tube parameters were selected automatically by the scanner depending on patient size. No intravenous contrast medium was used.

The image analysis for the pneumonia severity score was carried out by an automated lung opacity analysis program, “CT Pneumonia Analysis”, provided by Siemens Healthineers (Forchheim, Germany). The algorithm automatically detects and quantifies abnormal CT findings commonly present in lung infections, namely ground glass opacities (GGO) and consolidations. Based on 3D segmentations of lesions, lungs, and lobes, the algorithm measures volumes of lobes and lungs as well as high-opacity areas, including ground glass and consolidation. The AI program has previously been validated externally [23]. The severity of involvement (total opacity score) was measured as the ratio of the volume of high-opacity areas to that of total lung volume (Figure 2 and Figure 3).

### 2.3. Sample Size

In the main study, the recovery rate of COVID-19 patients with high-risk OSA was estimated to be around 80% within 28 days of hospitalization, compared to around 90% of patients with low-risk OSA [22]. Based on this assumption, and an 80% power (1-β) with the type I error (α = 0.05), the sample size for the hospitalized participants was calculated as 196 for the primary outcome. A separate power analysis was not performed for the current protocol, given that the aim of the study was to address the diagnostic utility of the AI-guided CT-based severity scores in a cross-sectional cohort of 221 patients, which is larger than the estimated sample size for the main study.

### 2.4. Statistics

For the statistical analysis, IBM SPSS Statistics for Windows, Version 26.0 (Armonk, NY, USA: IBM Corp.), was used. The Shapiro–Wilk test was conducted for the normality assumption. Descriptive statistics were shown using the mean and standard deviation (SD) for normally distributed variables, as well as the median (and interquartile range (IQR)) for non-normally distributed variables. Non-parametric statistical methods were conducted for values with a skewed distribution. Non-normally distributed groups were compared via a Mann–Whitney U test. The χ^2^ test (Fisher’s exact test where available) was used for categorical variables and expressed as observation counts (and percentages). The diagnostic parameters of the mBQ were computed across different SS cut-offs, including the diagnostic odds ratio, disease prevalence, sensitivity, specificity, negative likelihood ratio, negative predictive value, positive likelihood ratio, positive predictive value, and accuracy. An area under the receiver operating characteristic (ROC) curve analysis was used to assess the association between SSs and the need for supplemental oxygen as well as hospitalization and to predict the best SS cut-off value. Coefficients between 0.70 and 0.79 are generally regarded as acceptable; those between 0.80 and 0.89 are good; and those between 0.90 and 1.00 are considered excellent.

Univariate factors associated with the need for supplemental oxygen, as well as the need for hospitalization, were addressed by using a logistic regression analysis. In the multivariate analyses, the best SS cut-off value, age, sex, body mass index (BMI), diabetes, hypertension, and coronary artery disease were included in the model. Statistical significance was accepted when the two-sided *p*-value was lower than 0.05.

## 3. Results

The mean age of the patients was 52.6 ± 15.6 (range of 23 to 93) years, and the mean BMI was 27.7 ± 5.0 (range of 17.4 to 46.9) kg/m^2^.

The receiver operating characteristic (ROC) curve analysis of the association between SSs and the need for supplemental oxygen revealed a cut-off score of 2.65, with a sensitivity of 81% and a specificity of 56% (Figure 4).

As shown in Table 1, patients with SSs ≥ 2.65 were significantly older, had higher BMIs, and had more comorbidities in terms of hypertension and diabetes than those with lesser extents of pneumonic involvement, whereas other demographic and comorbid factors did not differ significantly. Of patients with SSs ≥ 2.65, 80% and 39% required hospitalization and supplemental oxygen, respectively, as compared to 55% and 11%, respectively, for patients with SSs < 2.65. The need for an ICU ward did not differ significantly between the two groups (Table 1).

Although not statistically significant, patients with an SS ≥ 2.65 tended to have more of a fever. These patients were more frequently dyspneic and had higher levels of C-reactive protein (CRP) than those with an SS < 2.65, whereas other clinical and laboratory parameters, including PCR positivity at admission, did not differ significantly (Table 2). All of the patients requiring supplemental oxygen (*n* = 11) in the group with an SS < 2.65 had a fever ≥ 38 °C and CRP ≥ 10 mg/dL at admission to the hospital.

SSs ≥ 2.65 predicted the need for supplemental oxygen with an almost four-fold OR, while other predictors were a high BMI (OR of 1.10), male sex (OR of 2.53), and hypertension (OR of 2.18). SSs ≥ 2.65 forecasted the need for hospitalization with an OR of 2.4, along with age (OR of 1.04) and BMI (OR of 0.92) (Table 3).

## 4. Discussion

The main conclusion of the current study is that AI-guided CT-based SSs can be used to predict the need for supplemental oxygen and hospitalization in patients with COVID-19 pneumonia. The ROC curve analysis of the association between SSs and the need for supplemental oxygen revealed a cut-off score of 2.65 on the CT images, with a sensitivity of 81% and a specificity of 56%. This cut-off level showed an almost four-fold risk increase in the need for supplemental oxygen and an almost 2.5-fold risk increase in the need for hospitalization in multivariate regression models adjusted for age, sex, BMI, diabetes, hypertension, and coronary artery disease.

Previously, Harmon et al. [24] conducted a study to evaluate the performance of deep learning algorithms in detecting COVID-19 pneumonia in chest CT scans using multinational datasets. The study showed that AI algorithms achieved up to 90.8% accuracy in distinguishing COVID-19 findings from those of other clinical entities, with 84% sensitivity and 93% specificity. The algorithms were trained on a diverse cohort of 1280 patients and evaluated on an independent test set. The results demonstrated the potential of AI to assist in the rapid evaluation of CT scans for COVID-19 diagnosis [24].

In a study by Li et al. [25], the researchers investigated the diagnostic utility of AI in distinguishing COVID-19 from community-acquired pneumonia in chest CT scans. In the study, the AI algorithm demonstrated its potential as a diagnostic tool by achieving an accuracy of 92.4% in distinguishing COVID-19 from community-acquired pneumonia [25].

Another study by Mei et al. [20] focused on the rapid diagnosis of COVID-19 via the use of AI. The researchers developed an AI system that could analyze chest CT scans and make a diagnosis in seconds. The system was trained on a dataset of 5335 cases, including 4,626 confirmed COVID-19 cases and 709 negative cases. The AI system quickly and accurately diagnosed COVID-19, achieving a high accuracy rate of 96% [20].

Kardos et al. [26] found that the sensitivity and specificity of a deep learning (DL)-based CT severity score for detecting COVID-19 pneumonia were 39.0% and 80.2%, respectively. In the test cohort, the positive predictive value was 68.0%, the negative predictive value was 55.0%, and the overall accuracy was 58.9% [26].

In the study conducted by Chrzan et al. [27], the authors aimed to evaluate the use of an AI analysis of high-resolution computed tomography (HRCT) images to predict the clinical severity of COVID-19 pneumonia. In the study, a percentage ground glass volume equal to or greater than the cut-off point of 29% was associated with an odds ratio (OR) of 7.53 for transfer to the ICU, while an absolute inflammation volume equal to or greater than the cut-off point of 831 cm3 was associated with an OR of 4.31 for in-hospital death [27].

Sezer et al. [28] investigated the prognostic role of an automated MDCT pneumonia analysis program as an early outcome predictor for COVID-19 pneumonia in 96 hospitalized patients. They divided the patients into two clinical groups based on their clinical status: good or bad clinical course. Total opacity scores (TOSs), intensive care unit (ICU) entry, and mortality rates were measured. TOS values were higher in patients older than 60 years and in patients with comorbidities. The authors concluded that the automated MDCT pneumonia analysis program could serve as a valuable tool for rapidly and reliably assessing the extent of COVID-19 pneumonia [28]; however, no specific thresholds were given for high vs. low TOSs.

In a retrospective study, Chaganti et al. [19] proposed a method to segment abnormalities associated with COVID-19 by using a non-contrast chest CT as an input. The method was based on a dataset of 9749 chest CTs and used deep learning as well as deep reinforcement learning. The method combined the metrics to determine the severity of lung and lobe involvement by measuring both the extent of COVID-19 abnormalities and the presence of high opacities. The method was evaluated on a dataset including CT studies of 200 COVID-19-confirmed patients, supporting the agreement between higher scores and clinical outcomes.

Thus, assessing the severity of COVID-19 pneumonia with the help of AI may have an important role in predicting the need for additional oxygen and hospitalization in patients with pneumonia. In particular, patients with more extensive lung involvement, as shown by AI, are likely to show more severe clinical courses and complications. These findings emphasize the potential role of AI in evaluating the risk factors of patients and supporting treatment decisions. Studies show that, in addition to clinical parameters, AI-based severity assessments can be important tools in better understanding the clinical courses of patients. These results reflect the increasing role and value of AI in medical diagnosis and management. They not only align with contemporary diagnostic trends but also have the potential to significantly improve patient care, research methodologies, and the overall quality of healthcare delivery [29,30].

Of note, 11 patients in the group with SSs < 2.65 requiring supplemental oxygen had a fever ≥ 38 °C and CRP ≥ 10 mg/dL at admission to the hospital. Given that the radiological changes may not be seen in the early phases of pneumonia, it should be underscored that the incorporation of radiology scans in conjunction with laboratory parameters is crucial in the diagnostic and initial clinical assessment process. In a meta-analysis, Stegeman et al. [21] included 21 studies with 14,126 COVID-19 patients and 56,585 non-COVID-19 patients in total, and evaluated 67 different laboratory tests. There was substantial heterogeneity between the tests, the threshold values, and the settings in which they were applied. For some tests a positive result was defined as a decrease compared to normal values, whereas it was defined as an increase compared to normal values in others, and for some tests both increases and decreases indicated test positivity. The authors observed that only three of the tests evaluated had a summary sensitivity and specificity over 50%, and they were an increase in interleukin-6, an increase in CRP, and a decrease in lymphocyte count. It was concluded that some tests may be specific indicators for inflammatory processes; however, none of the tests investigated were useful for accurately ruling in or ruling out COVID-19 on their own. The authors have also concluded that studies were carried out in specific hospitalized populations, and that non-hospital settings should be considered in future studies to evaluate how these tests would perform in people with milder symptoms [21]. Thus, routine laboratory tests cannot distinguish between COVID-19 and other diseases as the cause of infection, inflammation, or tissue damage, and none of the tests performed well enough to be a standalone diagnostic test for COVID-19, nor to prioritize patients for treatment. These tests may mainly be used to provide an overall picture about the health status of a patient, but the final COVID-19 diagnosis has to be made based on other tests, and in that context the synergy between radiology and laboratory data contributes to a more holistic and effective diagnostic approach, potentially improving patient outcomes and clinical decision making.

It should also be kept in mind that the price of medical treatments continues to rise due to an increasing population, aging, increasing disease prevalence, an increasing need of healthcare services, and increases in prices [30]. AI is already well known for its advantages in several healthcare applications, including the segmentation of lesions in images, speech recognition, smartphone personal assistants, and navigation. In a recent study, Khanna et al. [30] evaluated AI technology in the context of healthcare costs, particularly in the areas of diagnosis and treatment, and then compared it to traditional or non-AI-based approaches. They used PRISMA to select the best 200 studies for AI in healthcare, with a primary focus on cost reduction. They defined the diagnosis and treatment architectures, investigated their characteristics, and categorized the roles that AI plays in the diagnostic and therapeutic paradigms. They experimented with several combinations of different assumptions by integrating AI and then comparing it against conventional costs. In the model provided, the authors showed great cost savings when using AI tools in diagnosis and treatment. They concluded that the economics of AI can be improved by incorporating pruning, reducing AI bias, explainability, and regulatory approvals [30].

The strength of our study is that we applied AI-guided CT-based SSs in identifying a cut-off level for the prediction of the need for supplemental oxygen and hospitalization in a well-defined clinical cohort during the first outbreak of COVID-19 [22]. In a previous study, the AI system achieved an AUC of 0.92 and had equal sensitivity as compared to a senior thoracic radiologist in 2019 [20]. Remarkably, the AI system also advanced the detection of patients who tested positive for COVID-19 via PCR, who presented with normal CT scans, correctly recognizing 17 of 25 (68%) patients, whereas radiologists categorized all of these patients as COVID-19-negative. Our study is the first to integrate AI-guided CT-based SSs with short-term clinical outcomes, including the need for supplemental oxygen and hospitalization, and thus clinical management as well as decision making for triage and healthcare service prioritizations. This may also have cost–benefit aspects.

We should also acknowledge the limitations of our study. Firstly, the study was conducted on central areas of Istanbul, which might limit the generalizability of the findings to urban populations. Secondly, there might be unaccounted confounding variables that could influence the observed associations. Thirdly, the study focused on short-term outcomes, and the predictive ability of AI-guided SSs for long-term outcomes remains to be explored. Fourthly, expanding the training dataset and incorporating an external test set could significantly strengthen the study’s findings. Finally, the clinical significance of the observed associations between SSs and outcomes might require further investigation to be able to guide practical implications.

## 5. Conclusions

Our results clearly demonstrate that AI-guided CT-based severity scores can be used to predict the need for supplemental oxygen and hospitalization in patients with COVID-19 pneumonia. An SS threshold of 2.65 on CT images represents a meaningful cut-off point with 81% sensitivity and 56% specificity for predicting the need for supplemental oxygen. This threshold value has a special significance, is unique in the literature, and could serve as a powerful tool to accurately predict the need for supplemental oxygen and hospitalization of patients with COVID-19 pneumonia. These results demonstrate the growing importance of AI, which has the potential to go beyond clinical parameters to better understand the clinical course of patients and support treatment decisions.

## Figures and Tables

**Figure 1 jcm-12-07039-f001:**
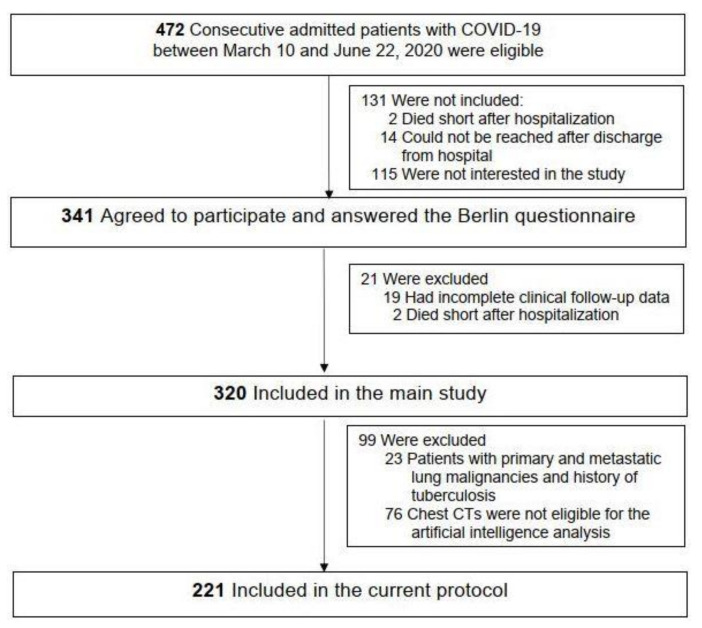
Flow of patients through the study. COVID-19 = coronavirus disease; CT = computed tomography.

**Figure 2 jcm-12-07039-f002:**
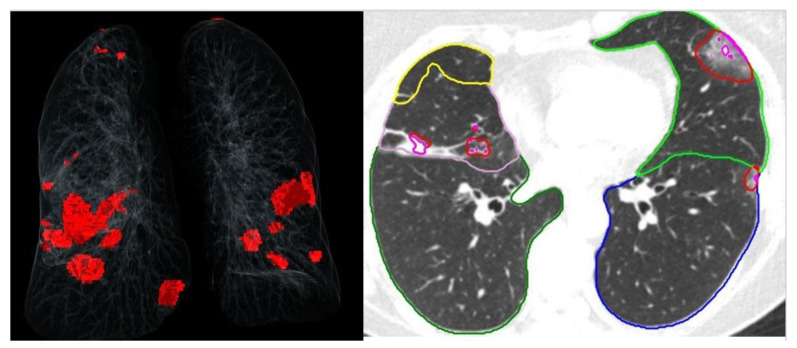
Chest tomography images and artificial intelligence calculations of a patient with a severity score of 1.1.

**Figure 3 jcm-12-07039-f003:**
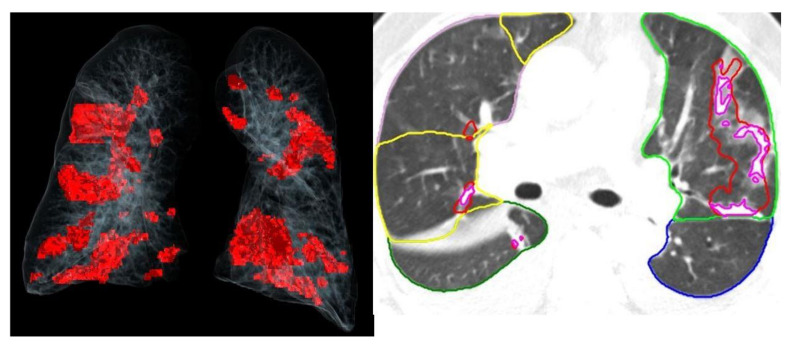
Chest tomography images and artificial intelligence calculations of a patient with a severity score of 4.8.

**Figure 4 jcm-12-07039-f004:**
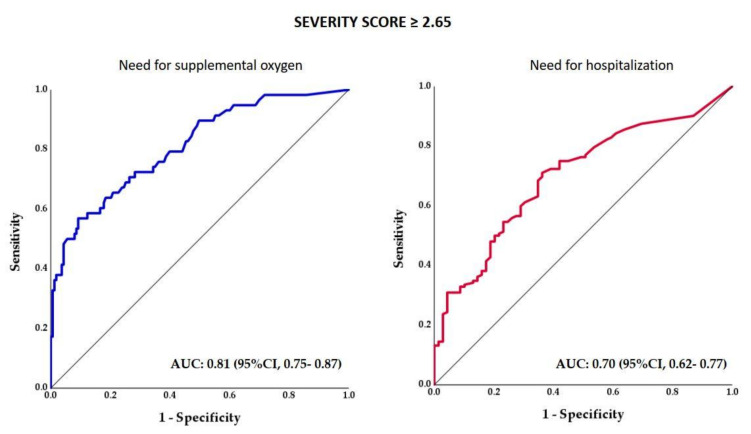
ROC analysis curve for patients requiring supplemental oxygen and hospitalization. Abbreviations: AUC = area under the curve; CI = confidence interval.

**Table 1 jcm-12-07039-t001:** Clinical characteristics of the groups dichotomized with respect to an SS of 2.65.

Variables	All Patients *n*= 221	SS < 2.65 *n* = 101	SS ≥ 2.65 *n* = 120	*p*-Value
SS	3.3 (0.5–9.4)	0.3 (0.1–1.2)	8.8 (4.4–16.4)	<0.001
Age, years	53.5 (40.6–62.5)	45.1 (35.8–58.0)	55.5 (49.5–66.6)	<0.001
Age ≥ 65 years	43 (19.5)	10 (9.9)	33 (27.5)	<0.001
Male sex	130 (58.8)	55 (54.5)	75 (62.5)	0.226
BMI, kg/m^2^	27.1 (24.7–30.9)	25.9 (23.1–30.1)	27.8 (25.5–31.29)	<0.001
Obesity	68 (30.8)	26 (11.8)	42 (35)	0.137
Current smoking	19 (9.2)	8 (8.3)	11 (9.9)	0.695
Diabetes mellitus	32 (14.5)	8 (7.9)	24 (20.0)	0.011
Hypertension	75 (33.9)	24 (23.8)	51 (42.5)	0.003
Congestive heart failure	3 (1.4)	0 (0)	3 (2.5)	0.252
Atrial fibrillation	3 (1.4)	0 (0)	3 (2.5)	0.252
Coronary artery disease	17 (7.7)	4 (4.0)	13 (10.8)	0.075
COPD	5 (2.3)	3 (3.0)	2 (1.7)	0.662
Asthma	5 (2.3)	2 (2.0)	3 (2.5)	1
ICU	14 (6.3)	4 (4.0)	10 (8.3)	0.268
Supplemental oxygen	58 (26.2)	11 (10.9)	47 (39.2)	<0.001
Hospitalization	152 (68.8)	56 (55.4)	96 (80.0)	<0.001

Definition of abbreviations: BMI = body mass index; COPD = chronic obstructive pulmonary disease; COVID-19 = coronavirus disease; ICU = intensive care unit; and SS = severity score. Continuous variables are expressed as medians (boundaries of interquartile ranges), and categorical variables are given in counts (percentages).

**Table 2 jcm-12-07039-t002:** Clinical and laboratory parameters of the groups dichotomized with respect to an SS of 2.65.

Parameter	All Patients *n* = 221	SS < 2.65 *n* = 101	SS ≥ 2.65 *n* = 120	*p*-Value
Fever ≥ 38 °C, *n* (%)	162 (73.3)	68 (67.3)	94 (78.3)	0.065
Cough, *n* (%)	137 (62.0)	62 (61.4)	75 (62.5)	0.865
Dyspnea, *n* (%)	63 (28.5)	20 (19.8)	43 (35.8)	0.009
Fatigue, *n* (%)	92 (41.6)	39 (38.6)	53 (44.2)	0.404
PCR positivity at admission, *n* (%)	103 (46.6)	50 (49.5)	53 (44.2)	0.428
Hemoglobin, g/dL	13.6 (12.6–14.5)	13.5 (12.7–14.5)	13.7 (12.3–14.4)	0.696
WBC, 10^9^/L	5.7 (4.5–7.6)	5.4 (4.3–7.6)	6.0 (4.6–7.6)	0.581
Platelets, 10^9^/L	203 (165–240)	211 (172–260)	198 (164–263)	0.462
CRP, mg/dL	25.8 (6.8–51.9)	10.3 (2.4–27.0)	39.9 (19.8–82.8)	<0.001

Definition of abbreviations: CRP = C-reactive protein; SS = severity score; and WBC = white blood cells. Continuous variables are expressed as medians (boundaries of interquartile ranges).

**Table 3 jcm-12-07039-t003:** Logistic regression results for the need for supplemental oxygen and hospitalization.

	Odds Ratio		95% CI for Odds Ratio		*p*–Value
			Lower	Upper	
**Need for** **Supplemental Oxygen**					
Severity score ≥ 2.65		3.98	1.80	8.78	<0.001
Age		1.02	0.99	1.05	0.13
BMI		1.10	1.02	1.19	0.01
Male sex		2.53	1.19	5.38	0.02
Hypertension		2.18	0.99	4.79	0.05
Diabetes		0.80	0.30	2.10	0.65
CAD		0.31	0.08	1.13	0.08
**Need for Hospitalization**					
Severity score ≥ 2.65		2.40	1.23	4.71	0.01
Age		1.04	1.01	1.06	0.01
BMI		0.92	0.86	0.98	0.02
Male sex		1.44	0.76	2.72	0.27
Hypertension		2.32	0.96	5.60	0.06
Diabetes		3.16	0.80	12.49	0.10
CAD		0.43	0.09	1.96	0.27

Definition of abbreviations: BMI = body mass index; CAD = coronary artery disease; and CI = confidence interval.

## Data Availability

Individual participant data reported in this article can be provided by contacting the corresponding author, yuksel.peker@lungall.gu.se.

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
