# Peer review of "The Diagnostic Utility of Artificial Intelligence-Guided Computed Tomography-Based Severity Scores for Predicting Short-Term Clinical Outcomes in Adults with COVID-19 Pneumonia"

_jcm, 2023, doi:10.3390/jcm12227039_

Round 1

Reviewer 1 Report

Comments and Suggestions for Authors

Review of the manuscript “The Diagnostic Utility of Artificial Intelligence Guided CT-based Severity Scores for Predicting Short-Term Clinical Outcomes in Adults with COVID-19 Pneumonia”

The objective of the study was to to examine the the relationship between AI-derived severity scores obtained from Computed Tomography (CT) scans and short-term clinical outcomes. Overall, the manuscript is well-structured and clearly written, with a good command of English and clear representation of the aim of the paper. Please consider the following issues:

Abstract: The abstract summarizes the major aspects of the entire paper, including the overall purpose of the study and the research problem, the basic design of the study, major findings and a brief conclusion.

Introduction: The introduction provides a clear and concise overview of the research problem, establishing its relevance in the field. The opening paragraph effectively captures the reader's attention by highlighting the significance of the topic. However:

§  The initial sections of the manuscript should prioritize the provision of precise and comprehensive data concerning the morbidity and mortality associated with SARS-CoV-2. Accurate characterization of these epidemiological parameters is essential for establishing the context and significance of the research findings presented in the manuscript: https://www.worldometers.info/coronavirus/

§  The incorporation of radiology scans in conjunction with laboratory parameters is of paramount importance in the diagnostic and initial clinical assessment process, PMID: 35976368, 36371787, 36972470, 33211319, and 32301746. In the context of this manuscript, the authors should underscore the significance of integrating radiology scans with laboratory data to enhance the diagnostic precision and refine the initial clinical assessment. Thus, the authors should elucidate how the synergy between radiology and laboratory data contributes to a more holistic and effective diagnostic approach, potentially improving patient outcomes and clinical decision-making. It is essential that the manuscript emphasizes the integration of these two modalities as a critical aspect of the diagnostic process.

Methods: The study methods are valid and reliable. The process of subject selection is clear. Variables are defined and measured appropriately. The authors employed appropriate statistical methods for the treatment of the data obtained. However:

§  Did author employe the power analysis?

§  Which method was used to determine whether sample data has been drawn from a normally distributed population?

§  Please provide explicit specifications for both the criteria that should be considered for inclusion and those that should be excluded.

§  I strongly recommend that the authors provide a comprehensive account of the ethical considerations and approvals related to their study.

Results: Text describes in detail the obtained values and their significance in drawing conclusions. Data are presented in an appropriate way.

§  Line 139: “...images with a sensitivity of % 81 and specificity of %56.” Please note that the percentage sign should be placed after the number, not in front of it. This ensures proper and conventional formatting in scientific writing.

Discussion: Relevance and importance of the obtained results are explained well in the discussion section. The results are discussed from multiple angles and placed into context without being overinterpreted. However:

§  Could you elaborate on the rationale behind the choice of a relatively small sample size for this study?

§  You mentioned that data was collected from two centers. Could you discuss any potential implications of this on the diversity and representativeness of the patient population in your study? Are there any specific characteristics or differences between these centers that might have influenced the results?

§  It was mentioned that the findings have not been externally validated using an independent dataset. Could you discuss the importance of external validation and any potential plans to conduct such validation in the future?

§  The focus of your research revolves around dynamic changes and AI. In the context of contemporary diagnostics, it is advisable to consider incorporating a discussion on the potential utility of AI algorithms, such as artificial neural networks, for tracking dynamic changes e.g. PMID: 37238309, 37515208, 33099829, 36554017. This could enrich your research by providing valuable insights into future perspectives. It not only aligns with contemporary diagnostic trends but also has the potential to significantly improve patient care, research methodologies, and the overall quality of healthcare delivery.

§  Your study focused on short-term outcomes. Are there any plans to investigate the predictive ability of AI-guided SS for long-term outcomes in pneumonia patients? If so, what are your expectations or hypotheses in this regard?

Conclusion: Conclusions answer the aims of the study and are fully supported by the results.

References: References come from reputable sources and all are cited properly throughout the manuscript.

This article tackles an important subject and presents valuable results. While there are areas that could be strengthened, the article lays a solid foundation for future research in the field. With the suggested improvements, this article has the potential to contribute significantly to our understanding of the relationship between AI-derived severity scores obtained from Computed Tomography (CT) scans and short-term clinical outcomes.

Reviewer 2 Report

Comments and Suggestions for Authors

1.      Introduction. The last paragraph of the introduction section is too short. To enhance the clarity and comprehensiveness of the manuscript, I recommend that the author extend this section. It should include a more elaborate introduction to the study's objectives and the main contributions. It would be beneficial for the author to provide an overview of the paper's organizational structure to guide the reader through the subsequent sections.

2.      Results. The authors mentioned that “The primary outcome was the need for supplemental oxygen and hospitalization over 28 days.” It is pertinent to inquire about the rationale behind selecting a 28-day timeframe as the standard. Could the authors explain whether this specific duration holds any clinical significance or pertinence in the context of the study objectives?

3.      Figures. Figures 1, 2, and 3 in the manuscript appear to lack the desired level of clarity. I recommend that the authors consider regenerating these figures with higher resolution to improve their visual quality.

4.      Results. The authors have employed a severity score threshold of >= 2.65 as a cutoff in their presentation of results. To enhance the context of this threshold selection, it is advisable that the authors provide a comprehensive presentation of the results from their ablation studies.

5.      Materials. Additional details regarding the training and testing data split, as well as a comprehensive description of the distribution of all characteristics across different cohorts, would significantly enhance the transparency and robustness of the study.

6.      Discussion. The discussion section features an extensive exposition of previous studies, but it lacks a direct comparison with the current findings. I suggest that the authors consider relocating some of the background information from the discussion to the introduction section and leave some methods in the discussion for a more comprehensive comparison of their results with prior research. Specifically, the authors should emphasize the advantages of their method in making the prediction and address the significance of their findings within the broader research landscape.

7.      Discussion. The inclusion of both successful and unsuccessful cases, along with a detailed explanation of why some predictions failed, would be a valuable addition to the manuscript.

8.      Discussion. The limitations section of the manuscript appears rather extensive. It would be beneficial for the authors to consider addressing some of these limitations more directly. For instance, expanding the training dataset and incorporating an external test set could significantly strengthen the study's findings and bolster their trustworthiness.

9.      The language should be further polished for this manuscript.

Comments on the Quality of English Language

The language should be further polished for this manuscript.

Reviewer 3 Report

Comments and Suggestions for Authors

The authors in their manuscript tried to use the AI approach to predict COVID-19 Pneumonia. So, regarding the the study goal, there are some minor issue for inproving the quality of work: 

1. The grammatical RV is necessary.

2. The methodology must be revised from the related guidelines.

3. The discussion need more related studies on subject for better understanding the study outcome.

4.The limitation of the AI approach in the study, from different aspect, should be mention in the end of duscussion.

Comments on the Quality of English Language

Minor grammatical RV.

Reviewer 4 Report

Comments and Suggestions for Authors

This is a very important study that add a value. I have only the following minor comments:

1. Line 31-32: write the % after the numbers.

2. Line 53: correct the references (the numbers)

3. Line 66: CT abbreviation is mentioned before (Line 53 without abbreviation), write it in full only in the first appearance.

4. Please, try to include more related cited references.

Round 2

Reviewer 1 Report

Comments and Suggestions for Authors

Dear Authors,

I have thoroughly reviewed the revised version of your manuscript and appreciate the comprehensive analysis and insights you have provided.

However, I noticed that this specific topic could benefit from a more comprehensive examination of recent advancements in the area. Considering the recent developments, it might be valuable to include insights from recent studies and references, such as those I have clearly mentioned in the previous report, which provide additional context for the discussion you presented.

Ensuring that the latest research in this field is appropriately represented will enhance the overall impact and credibility of your findings. I encourage you to consider incorporating these insights into your manuscript for a more comprehensive and well-rounded discussion of the topic.

Thank you for your attention to this matter. I look forward to seeing the revised version of your manuscript.
